# Neural Networks as Universal Finite-State Machines: A Constructive Feedforward Simulation Framework for NFAs

## Abstract

We present a formal and constructive simulation framework for nondeterministic finite automata (NFAs) using standard feedforward neural networks. Unlike prior approaches that rely on recurrent architectures or post hoc extraction methods, our formulation symbolically encodes automaton states as binary vectors, transitions as sparse matrix transformations, and nondeterministic branching—including $\varepsilon$-closures—as compositions of shared thresholded updates. We prove that every regular language can be recognized exactly by a depth-unrolled feedforward network with shared parameters, independent of input length. Our construction yields not only formal equivalence between NFAs and neural networks, but also practical trainability: we demonstrate that these networks can learn NFA acceptance behavior through gradient descent using standard supervised data. Extensive experiments validate all theoretical results, achieving perfect or near-perfect agreement on acceptance, state propagation, and closure dynamics. This work establishes a new bridge between symbolic automata theory and modern neural architectures, showing that feedforward networks can perform precise, interpretable, and trainable symbolic computation.

## 1 Introduction

The relationship between symbolic computation and neural networks has long fascinated both the theoretical computer science and machine learning communities. Finite automata Rabin & Scott (1959); Hopcroft et al. (2006); Sipser (1996) are among the most fundamental models of computation, capturing the essence of regular languages, while modern neural architectures LeCun et al. (2015); Goodfellow et al. (2016); Nair & Hinton (2010); Glorot et al. (2011); Lecun et al. (1998); Hochreiter & Schmidhuber (1997); Vaswani et al. (2017) are the cornerstone of deep learning systems. Despite their seemingly disparate origins, a growing body of research has sought to reconcile these paradigms by simulating automata within neural frameworks (Giles et al., 1992; Weiss et al., 2018; Graves et al., 2014).

Early attempts primarily employed recurrent networks (RNNs) to approximate deterministic finite automata (DFAs), leveraging their sequential dynamics to process string inputs (Giles et al., 1991; Korsky & Berwick, 2019). However, these approaches often required complex training, lacked interpretability, and provided no symbolic guarantees. More recent work has explored extracting automata from trained networks using queries and counterexamples (Weiss et al., 2018), or designing architectures with embedded stack or memory structures (DuSell & Chiang, 2022; Reed & de Freitas, 2016), but symbolic fidelity remained elusive.

In this paper, we introduce the first formal and constructive simulation framework for nondeterministic finite automata (NFAs) using feedforward neural networks with shared parameters. Unlike previous approaches that rely on recurrent architectures or extractive post hoc methods, our formulation symbolically encodes NFA states as binary vectors, transitions as sparse symbolic matrices, and nondeterministic branching—including $\varepsilon$-closures—as thresholded matrix compositions over shared transition matrices. We prove that every regular language accepted by an $n$-state NFA can be exactly simulated by our proposed framework, without requiring recurrence, memory, or approximation mechanisms. Our work builds upon but significantly extends previous studies on neural-symbolic systems (Bhattamishra et al., 2020; Merrill et al., 2020; Ergen & Grillo, 2024; Graves et al., 2014)

The key contributions of this work are as follows.

1. We present the first exact simulation of nondeterministic finite automata (NFAs) using feedforward networks, encoding states as binary vectors and transitions as sparse symbolic matrices.

2. We establish a constructive equivalence between symbolic feedforward networks and NFAs by proving that every regular language can be recognized by such a network. The model uses $\mathcal{O}(kn^2)$ parameters, independent of input length, enabling scalable simulation of arbitrarily long strings.

3. We show that $\varepsilon$-closure can be computed via iterative matrix-based thresholded updates, converging in at most $n$ steps.

4. We demonstrate that feedforward NFA simulator networks are learnable via gradient descent, achieving high accuracy in replicating NFA acceptance behavior using standard training on labeled data.

5. We validate all theoretical results across multiple NFA configurations, showing perfect or near-perfect agreement on state propagation, acceptance, and symbolic equivalence.

Together, these results provide the first formal bridge between classical automata theory and deep learning. This paper uses a structure with theorems, lemmas, and corollaries, similar to prior work Zhang et al. (2015); Hanin & Sellke (2018); Neyshabur et al. (2015); Dhayalkar (2025c;b;a); Stogin et al. (2024).

## 2 Related Work

This work builds on prior research at the intersection of automata theory, neural networks, and symbolic computation, and highlights the novelty of our approach relative to existing efforts.

**Automata Simulation by Neural Networks:** The idea of using neural networks to model automata has received significant attention. Early work explored the approximation of regular languages using RNNs Giles et al. (1992), including techniques for training second-order networks to accept deterministic finite automata (DFAs) Giles et al. (1991). More recent work has focused on extracting symbolic automata from trained RNNs using queries and counterexamples Weiss et al. (2018) or differentiable memory models such as Neural Turing Machines Graves et al. (2014). However, these models typically rely on recurrent or sequential architectures with opaque representations.

**Symbolic Interpretability of Neural Models:** Efforts to interpret neural networks symbolically have grown with the popularity of neural-symbolic learning. Notable examples include differentiable pushdown automata DuSell & Chiang (2022), differentiable interpreters Reed & de Freitas (2016), and formal grammars embedded into transformer and RNN architectures Bhattamishra et al. (2020); Merrill et al. (2020). These approaches aim to bridge symbolic logic and differentiable computation but often involve approximate, heuristic, or complex inductive mechanisms. Our work provides an exact, symbolic simulation of NFAs using only feedforward layers—no recurrence, memory modules, or training heuristics are required.

**Feedforward Networks and Formal Language Theory:** The theoretical properties of neural networks with thresholded linear operations have been explored in the context of expressivity and piecewise linearity Nair & Hinton (2010); Goodfellow et al. (2016). Recent studies have investigated how such architectures can represent formal structures Dhayalkar (2025b); Ergen & Grillo (2024). However, no prior work has formally characterized the class of regular languages as exactly simulatable by feedforward networks. In contrast, we show that every NFA (and thus every regular language) can be encoded via feedforward layers, and that subset construction, $\varepsilon$-closures, learnability, and acceptance can be realized as exact compositions of symbolic matrix updates and thresholding operations.

**Learning Formal Languages with Neural Networks:** Supervised learning of regular languages using neural networks has also been explored Butoi et al. (2025). However, these works treat neural networks as black-box approximators, aiming for empirical accuracy without structural guarantees. In contrast, our

Theorem 5.1 shows that when trained with a labeled dataset, the symbolic feedfoward network not only approximates but provably converges to exact simulators of the underlying NFA, achieving both symbolic fidelity and empirical learnability within a standard gradient descent framework.

**Equivalence Results:** While neural networks are known as universal function approximators, their symbolic equivalence to classical computational models remains underexplored. Theorem 4.10 establishes a formal equivalence between NFAs and feedforward networks for language recognition. In contrast to prior work on deterministic finite automata (DFAs) simulated by feedforward networks Dhayalkar (2025c), which unroll DFA transitions into ReLU or threshold layers, our results extend to *nondeterministic* automata—including $\varepsilon$-transitions and closure dynamics—and establish exact recognizability of all regular languages using feedfoward networks. To our knowledge, this is the first symbolic and constructive theorem connecting finite-state automata and standard feedforward neural networks.

## 3 Preliminaries and Formal Definitions

In this section, we formally define the key mathematical concepts used throughout the paper.

**Definition 3.1** (Non-Deterministic Finite Automaton (NFA)). An NFA is a tuple $\mathcal{A} = (Q, \Sigma, \delta, q_0, F)$ where:

- $Q$ is a finite set of states with $|Q| = n$,
- $\Sigma$ is the input alphabet with $|\Sigma| = k$,
- $\delta : Q \times (\Sigma \cup \{\varepsilon\}) \to 2^Q$ is the transition function (possibly including $\varepsilon$-transitions),
- $q_0 \in Q$ is the initial state, and
- $F \subseteq Q$ is the set of accepting states.

The transition function $\delta$ may include transitions labeled with $\varepsilon$, the empty string. These are called $\varepsilon$-*transitions* and allow the automaton to move between states without consuming an input symbol. A string $x \in \Sigma^*$ of length $L$ is accepted by $\mathcal{A}$ if there exists a sequence of transitions from the initial state $q_0$ to some state $q \in F$ that processes all symbols of $x$.

**Definition 3.2** (Language Accepted by an NFA). Given an NFA $\mathcal{A} = (Q, \Sigma, \delta, q_0, F)$, the language accepted by $\mathcal{A}$, denoted $\mathcal{L}(\mathcal{A})$, is defined as:

$$\mathcal{L}(\mathcal{A}) := \{x \in \Sigma^* \mid \text{there exists a run of } \mathcal{A} \text{ on } x \text{ that ends in a state } q \in F\}.$$

That is, $\mathcal{L}(\mathcal{A})$ contains all strings over the alphabet $\Sigma$ that are accepted by the automaton.

**Definition 3.3** (Feedforward Network with Binary Threshold Activation). A feedforward network with binary threshold activation and depth $D$ (i.e. $D$ layers) is a function $f_\theta : \mathbb{R}^d \to \mathbb{R}$ defined as a composition of thresholded linear transformations:

$$f_\theta(x) = W_D \cdot \sigma(W_{D-1} \cdot \sigma(\cdots \sigma(W_1 x + b_1) \cdots) + b_{D-1}) + b_D,$$

where each $W_i \in \mathbb{R}^{d_i \times d_{i-1}}$, $b_i \in \mathbb{R}^{d_i}$ are trainable parameters, and $\sigma(z) = \mathbf{1}_{[z>0]}$ is an elementwise thresholding nonlinearity (optionally relaxed during training).

**Definition 3.4** ($\varepsilon$-Closure). Given a state $q \in Q$, the $\varepsilon$-closure of $q$, denoted $\mathrm{cl}_\varepsilon(q)$, is the set of states reachable from $q$ via a sequence of zero or more $\varepsilon$-transitions. For a set of states $S \subseteq Q$, we define: $\mathrm{cl}_\varepsilon(S) = \bigcup_{q \in S} \mathrm{cl}_\varepsilon(q)$.

**Definition 3.5** (Symbolic Transition Matrix). For each $x_t \in \Sigma \cup \{\varepsilon\}$, define the symbolic transition matrix $T^{x_t} \in \{0,1\}^{n \times n}$ such that $T^{x_t}_{ij} = 1 \iff q_j \in \delta(q_i, x_t)$. It captures all possible transitions labeled by $x_t$.

**Definition 3.6** (Simulation of an NFA by a Feedforward Network). A feedfoward network $f_\theta$ is said to simulate an NFA $\mathcal{A} = (Q, \Sigma, \delta, q_0, F)$ if for every input string $x \in \Sigma^*$,

$$f_\theta(x) = 1 \iff x \in \mathcal{L}(\mathcal{A}),$$

i.e., the network accepts a string if and only if the automaton does.

**Definition 3.7** (Equivalence Between Feedforward Network and NFA). A feedfoward network $f_\theta$ and an NFA $\mathcal{A}$ are said to be equivalent if they recognize the same language:

$$\forall x \in \Sigma^*, \quad f_\theta(x) = 1 \iff \mathcal{A} \text{ accepts } x.$$

## 4   Theoretical Framework

**Proposition 4.1** (Binary State Vector Representation)**.** *Let $Q = \{q_1, \ldots, q_n\}$ be the set of states of an NFA. Define the one-hot indicator vector $e_i \in \{0, 1\}^n$ such that $[e_i]_j = \mathbf{1}_{[j=i]}$. Then any subset of states $S \subseteq Q$ can be represented as a binary vector $s \in \{0, 1\}^n$ defined by:*

$$s = \sum_{q_i \in S} e_i.$$

*The vector $s$ is called the* state vector *encoding of $S$. For any such $s$ and symbolic transition matrix $T^{x_t} \in \{0, 1\}^{n \times n}$, the updated vector*

$$s' = \mathbf{1}_{[T^{x_t} s > 0]}$$

*(where the indicator is applied elementwise) encodes the set of states reachable from $S$ under the transition relation defined by $T^{x_t}$, where $T^{x_t}$ encodes transitions on input symbol $x_t \in \Sigma$.*

*Proof.* The complete proof is provided in Appendix A.1.

Explanation: This proposition formalizes how the internal configuration of an NFA, typically represented as a subset of active states, can be embedded into a binary vector and updated via a matrix multiplication followed by a binary threshold. Each entry of $s_t$ reflects the activity of state $q_i$ at time $t$, and the transition matrix $T^{x_t}$ encodes the NFA's transition function on input symbol $x_t$ at time $t$.

Interpretation and Insight: This construction shows that a forward pass through the layer corresponds precisely to the propagation of nondeterministic state activations in an NFA, thereby simulating one time step of computation within a neural substrate.

Note on $\varepsilon$-Transitions: This result assumes $\varepsilon$-transitions are disabled. Handling $\varepsilon$-closures requires additional logic and is addressed in Lemma 4.6. This simplification aids theory without loss of generality.

**Remark 4.2** (On Binary Thresholding for Interpretability)**.** In Proposition 4.1, we assume a binary thresholding function to ensure interpretability of the state vector representation. This activation produces vectors in $\{0, 1\}^n$, making it easy to directly inspect which automaton states are active after each transition. The binary output aligns closely with the classical semantics of finite automata, where states are either active or inactive at each step.

For the remainder of this paper, we adopt binary thresholding as the default activation function in theoretical constructions, unless stated otherwise. However, this choice is not essential for correctness. In practice, when training the feedforward network with labeled data (explained in detail in Theorem 5.1), the thresholding function can be replaced with ReLU, sigmoid, or even omitted altogether to facilitate differentiable optimization. Experimental results comparing these alternatives are provided in Section 6.7.

**Remark 4.3.** Although the state vector $s$ is initially binary and lies in $\{0, 1\}^n$, its transformation through the transition matrix $T^{x_t} \in \{0, 1\}^{n \times n}$ results in a vector $T^{x_t} s \in \mathbb{N}^n$, where each entry counts the number of activations received by a state. The binary threshold function $\mathbf{1}_{[T^{x_t} s > 0]}$ then converts this into a new binary state vector indicating which states are reachable.

In practice, particularly during training or when generalizing to soft-symbolic computation, intermediate values of $T^{x_t} s$ may lie in $\mathbb{R}^n$, especially if the transition matrices are parameterized and learned. In such cases, the thresholding function $\mathbf{1}_{[z>0]}$ can be approximated or replaced with another activation function like ReLU, and the activation vector may lie in $\mathbb{R}^n_{\geq 0}$. Nevertheless, the simulation remains sound as long as downstream interpretation relies only on identifying positive entries—i.e., which states are considered active—rather than their exact values.

**Remark 4.4** (Parallel Path Tracking)**.** Binary-threshold state vector updates naturally support the parallel tracking of multiple computational paths in NFAs. Given a state vector $s_t \in \{0, 1\}^n$ representing a subset of active states at time $t$, and a symbol-conditioned transition matrix $T^{x_t} \in \{0, 1\}^{n \times n}$, the update rule $s_{t+1} = \mathbf{1}_{[T^{x_t} s_t > 0]}$ computes the union of next reachable states from all active states—propagating all nondeterministic branches in one matrix-vector product.

Constructive Interpretation: For each state $q_i$, the next activation is:

$$[s_{t+1}]_i = \mathbf{1}_{\left[\sum_{j=1}^n T_{ij}^{x_t} \cdot [s_t]_j > 0\right]}$$

If the sum is positive, state $q_i$ is reachable from some active state at time $t$ under input symbol $x_t$; otherwise, it remains inactive. Thus, the entries of $s_{t+1}$ directly identify the reachable states at time $t+1$. As mentioned in Remark 4.3, intermediate computation of $\sum_{j=1}^n T_{ij}^{x_t} \cdot [s_t]_j$ may lie in $\mathbb{R}$ during training with soft parameters.

Insight: Each row of the transition matrix $T^{x_t}$ encodes all possible transitions for a given input, and the threshold function propagates all reachable states in parallel. This mirrors the subset construction in automata theory, but without explicitly computing power sets—the state vectors implicitly track all active branches.

Prior Work: Traditional methods for simulating NFAs often rely on sequential or recursive processing Korsky & Berwick (2019). This work introduces a novel perspective by showing how feedforward networks can capture all possible transitions in parallel, enabling efficient automata simulation within neural architectures.

**Theorem 4.5** (Subset Construction using Matrix-based Thresholded Updates). *Let $\mathcal{A} = (Q, \Sigma, \delta, q_0, F)$ be an NFA with $|Q| = n$. For an input string $x = x_1 x_2 \cdots x_L \in \Sigma^L$, let $\{T^{x_t}\}_{t=1}^L$ be the corresponding sequence of transition matrices. Let $s_0 \in \{0, 1\}^n$ denote the one-hot vector for start state $q_0$. Then, the sequence of thresholded updates*

$$s_t = \mathbf{1}_{[T^{x_t} s_{t-1} > 0]}, \quad \text{for } t = 1, \ldots, L$$

*computes the active state set at time $t$, i.e., $s_t \in \{0, 1\}^n$ is the binary indicator vector for the subset $S_t \subseteq Q$ reachable from $q_0$ by consuming prefix $x_1 \cdots x_t$. The final vector $s_L$ thus represents the subset construction $\delta(q_0, x_1 \cdots x_L)$.*

*Proof.* The complete proof is provided in Appendix A.2.

Explanation: This theorem formalizes the compositional structure of nondeterministic computation using matrix-based thresholded updates. Each application of the update rule $s_t = \mathbf{1}_{[T^{x_t} s_{t-1} > 0]}$ simulates one transition step of the NFA. By chaining these updates, a feedforward network using only matrix multiplication and thresholding can simulate the entire computation over a finite-length string input. The final output vector $s_L$ encodes the subset of states reached by the NFA on input $x$.

Interpretation and Insight: The construction mirrors the classical subset construction used to determinize NFAs, but instead of explicitly enumerating all subsets of $Q$, it represents and evolves the active subset implicitly using vector encodings and thresholded linear updates.

Depth of the network: The depth of this network grows linearly with the input length. However, it is important to emphasize that **the number of distinct parameters in the network is small and independent of the input length**. Specifically, each symbol $x \in \Sigma$ has an associated fixed input-length-independent transition matrix $T^x \in \{0, 1\}^{n \times n}$, and there is a single $\varepsilon$-transition matrix $T^\varepsilon \in \{0, 1\}^{n \times n}$. These matrices are reused across all time steps. This is explained more in detail in Theorem 4.7 and Proposition 4.9.

Relation to Prior Work: Classical neural simulations of NFAs typically use recurrent architectures Giles et al. (1992); Weiss et al. (2018), treating transitions as emergent and lacking symbolic guarantees. In contrast, our approach uses stacked threshold-based updates to explicitly simulate subset construction, preserving interpretability and exact automata semantics. To our knowledge, this is the first formal embedding of the full subset construction into standard feedforward networks without recurrence or memory.

**Lemma 4.6** ($\varepsilon$-Closure via Matrix-Based Thresholded Propagation). *Let $\mathcal{A} = (Q, \Sigma \cup \{\varepsilon\}, \delta, q_0, F)$ be an NFA with $|Q| = n$, and let $T^\varepsilon \in \{0, 1\}^{n \times n}$ be the transition matrix corresponding to $\varepsilon$-transitions. Then the iterative update*

$$s^{(k+1)} = \mathbf{1}_{[T^\varepsilon s^{(k)} > 0]}, \quad s^{(0)} = s_t,$$

*converges in at most $n$ steps to the $\varepsilon$-closure of $s_t$, i.e., the smallest superset of active states reachable via zero or more $\varepsilon$-transitions.*

*Proof.* The complete proof is provided in Appendix A.3.

Explanation: In classical automata theory, the $\varepsilon$-closure of a set of NFA states is the set of all states reachable via paths composed solely of $\varepsilon$-transitions. This lemma shows that the iterative rule $s^{(k+1)} = \mathbf{1}_{[T^\varepsilon s^{(k)} > 0]}$ acts as a symbolic fixed-point process that computes this closure. Beginning from a binary indicator vector $s^{(0)}$ for some state subset $S_0 \subseteq Q$, repeated application of $T^\varepsilon$ followed by thresholding converges to the full $\varepsilon$-closure of $S_0$. The thresholding ensures that only reachable states are activated, while the matrix product propagates transitions. Since each transition matrix is binary and $|Q| = n$, convergence occurs in at most $n$ steps, after which no new states are added.

Clarification: The matrix $T^\varepsilon$ need not be nilpotent—$\varepsilon$-transitions can form cycles. However, because the update rule is monotonic and the state space is finite, each new state added remains active, and at most $n$ states can be activated. Therefore, convergence is guaranteed in at most $n$ steps, regardless of whether the $\varepsilon$-transition graph is cyclic or acyclic.

Insight and Relation to Prior Work: Classical algorithms for computing $\varepsilon$-closures—such as depth-first and breadth-first search—are well-established in automata theory Aho & Hopcroft (1974). Our approach departs from these by casting the closure operation as a symbolic, differentiable iteration based on repeated matrix multiplications followed by thresholding, removing the need for control flow or recursion. To our knowledge, this is the first framework to encode $\varepsilon$-closure as a convergent fixed-point iteration that integrates cleanly into symbolic or neural computation.

**Theorem 4.7** (Simulation of NFAs via Feedforward Networks). *Let $\mathcal{A} = (Q, \Sigma \cup \{\varepsilon\}, \delta, q_0, F)$ be an NFA with $|Q| = n$ states. Then, for any input string $x = x_1 x_2 \ldots x_L \in \Sigma^*$, there exists a computation composed of $L$ alternating layers of symbol-transition propagation and $\varepsilon$-closure operations such that the final binary state vector $s_L \in \{0,1\}^n$ encodes the reachable states of $\mathcal{A}$ after processing $x$. Furthermore, acceptance ($x \in \mathcal{L}(\mathcal{A})$) is equivalent to $\langle s_L, \mathbf{1}_F \rangle > 0$, where $\mathbf{1}_F \in \{0,1\}^n$ is the indicator vector for the accepting state set $F$.*

*Formally, the following recursion simulates the NFA using thresholded matrix updates:*

$$s_0 = \mathbf{1}_{[(T^\varepsilon)^n s_{q_0} > 0]},$$
$$s_t = \mathbf{1}_{[(T^\varepsilon)^n \cdot \mathbf{1}_{[T^{x_t} \cdot s_{t-1} > 0]} > 0]}, \quad for\ t = 1, \ldots, L,$$
$$Accept(x) = True \iff \langle s_L, \mathbf{1}_F \rangle > 0.$$

*where $T^\varepsilon$ is the $\varepsilon$-transition matrix, and $(T^\varepsilon)^n$ denotes repeated application until convergence (at most $n$ steps) as mentioned in Lemma 4.6.*

*Proof.* The complete proof is provided in Appendix A.4.

Explanation: The simulation proceeds in phases: (1) initializing the active state vector using the $\varepsilon$-closure of the start state, (2) alternating input-symbol transitions and $\varepsilon$-closures at each step, and (3) checking whether any accepting state becomes active. Each operation—symbol-based transition or $\varepsilon$-closure—is implemented using a sparse matrix-vector multiplication followed by elementwise thresholding. This yields a purely symbolic yet compositional computation model.

Structure: Each input symbol $x_t$ triggers a symbolic matrix transition via $T^{x_t}$, followed by $\varepsilon$-closure computed using $T^\varepsilon$. This implies the depth of the network is $O(L)$, with width $n$ (equal to the number of NFA states). Importantly, **the number of distinct parameters in the network is small and independent of the input length**. Specifically, each symbol $x \in \Sigma$ has an associated transition matrix $T^x$, and there is a single $\varepsilon$-transition matrix $T^\varepsilon$, both of which are reused at every step. Thus, even though the network may have a large depth proportional to the input length $L$, **all layers share a small set of symbolic parameters** corresponding to the underlying automaton structure. This is explained more in detail in Proposition 4.9

Insight and Contribution: This theorem completes the simulation pipeline: from initialization, through alternating symbolic and $\varepsilon$ transitions, to final acceptance via an inner product. It connects symbolic automata processing to continuous differentiable computation while preserving interpretability—each layer mimics a specific automaton step.

Comparison to Prior Work: While classical automata simulation relies on pointer-based state transitions and recursion, this model provides a linear-algebraic, fully differentiable view. Unlike RNN-based embeddings,

this construction retains full symbolic fidelity to NFA semantics while staying within the architecture of standard feedforward networks. To the best of our knowledge, this is the first such construction for NFAs.

## 4.1 Encoding Input Strings into the Feedforward Network

A key aspect of our construction is how an input string $x = x_1 x_2 \cdots x_L \in \Sigma^*$ is provided to the feedfoward network. Unlike conventional neural networks where the input string is embedded into a single vector and passed into the network as a feature input, our construction decouples the string from the network architecture. Instead, the string determines the sequence of symbolic transition matrices that are applied during the forward pass.

Transition Matrix Selection: For every symbol $x \in \Sigma$, the model maintains a corresponding symbolic transition matrix $T^x$ that encodes all transitions in the automaton labeled with $x$. These matrices can either be:

- Fixed (Symbolic): Directly encoded from the known transition function $\delta$ of the target NFA, as described in Definition 3.5.
- Learned: Parameterized and initialized randomly, then optimized via gradient descent using labeled string examples, as shown in Theorem 5.1.

Encoding the String: To process the input string $x = x_1 x_2 \ldots x_L$, the feedfoward network does not take $x$ as a conventional input vector. Instead, the symbols in the input string act as control instructions that determine which transition matrices are applied at each layer. Specifically, at each step $t$, the symbol $x_t$ selects the matrix $T^{x_t}$, and the state vector is updated using thresholding as described in Proposition 4.1:

$$s_t = \mathbf{1}_{[T^{x_t} \cdot s_{t-1} > 0]}$$

This update corresponds to the propagation of active NFA states under symbol $x_t$ from step $t-1$ to step $t$. When $\varepsilon$-transitions are present, this is followed by closure computation as described in Lemma 4.6:

$$s_t = \mathbf{1}_{[(T^\varepsilon)^n \cdot \mathbf{1}_{[T^{x_t} \cdot s_{t-1} > 0]} > 0]}$$

where $T^\varepsilon$ is the $\varepsilon$-transition matrix, and $(T^\varepsilon)^n$ denotes repeated application of thresholded propagation until convergence (at most $n$ steps).

Symbol-Driven Matrix Selection: Each symbol $x_t$ in the input string $x = x_1 x_2 \ldots x_L$ acts as a selector that determines which transition matrix $T^{x_t}$ is applied at step $t$ of the forward pass. Rather than embedding the entire input string into a continuous vector, our construction interprets the string as a sequence of symbolic instructions that dynamically select matrices from a fixed set $\{T^x\}_{x \in \Sigma} \cup \{T^\varepsilon\}$. These matrices define the network's computation path. This mechanism mirrors the transition semantics of NFA, where each symbol determines state updates via the corresponding transition relation, and stands in contrast to typical neural sequence models that rely on learned embeddings or positional encodings.

An example of encoding input strings into the network is provided in Appendix A.5

## 4.2 Regular Language Recognition, Parameter Efficiency and Equivalence

**Remark 4.8** (Regular Language Recognition). Every regular language can be recognized exactly by constructing a feedforward network as described in Theorem 4.7. Specifically, for any NFA $\mathcal{A}$, the constructed network simulates all state transitions—including $\varepsilon$-closures—and determines acceptance via a readout layer. The resulting architecture performs symbolic computation using thresholded matrix operations and shared transition matrices, thereby serving as an exact recognizer for the regular language defined by $\mathcal{A}$.

**Proposition 4.9** (Parameter Efficiency of Feedforward Automata Simulators with Symbolic and $\varepsilon$-Transitions). *Let $\mathcal{A} = (Q, \Sigma \cup \{\varepsilon\}, \delta, q_0, F)$ be an NFA with $|Q| = n$ states, $|\Sigma| = k$ input symbols, and $\varepsilon$-transitions allowed. Construct a feedforward network $f_\theta$ that simulates $\mathcal{A}$ according to Theorem 4.7, using a transition matrix $T^x$ for each symbol $x \in \Sigma$, and a transition matrix $T^\varepsilon$ encoding $\varepsilon$-transitions.*

*Then the following holds:*

1. *The total number of trainable parameters is bounded by $\mathcal{O}(kn^2 + n^2) = \mathcal{O}(kn^2)$.*

2. *This parameter count (and hence the number of layers) is independent of the input length $L$.*

*Proof.* The complete proof is provided in Appendix A.6.

Explanation: This proposition highlights a key structural property: the model's parameter count is independent of input length. Although it simulates computations over arbitrarily long strings, the feedforward network reuses a fixed set of symbolic transition matrices at each step—mirroring how NFAs operate with a fixed transition function regardless of string length.

Insight and Implication: Unlike RNNs or transformers, where model complexity often grows with sequence length or attention span, our model decouples representational capacity from input length. This enables scalable regular language recognition using bounded memory and compute, keeping training, inference, and memory usage tractable even for long inputs.

Relation to Prior Work: Prior work on neural-symbolic learning has explored recognizing regular languages using RNNs or transformers Weiss et al. (2018); Butoi et al. (2025); Liu et al. (2023), but often entangles symbolic representation with recurrence or depth. Our approach cleanly separates symbolic operators (transition matrices) from dynamic unrolling, enabling both theoretical analysis and efficient implementation.

**Theorem 4.10** (Equivalence between Feedforward Networks and NFAs). *Let $\mathcal{L} \subseteq \Sigma^*$ be any regular language. Then:*

1. ***(Forward Direction)*** *There exists a symbolic feedforward network $f_\theta$ with:*

   - *transition matrices $\{T^x\}_{x \in \Sigma}$ and $T^\varepsilon$ representing the NFA structure,*
   - *weight sharing across time steps,*
   - *width $\mathcal{O}(n)$, where $n$ is the number of NFA states,*

   *such that for any input string $x = x_1 x_2 \ldots x_L \in \Sigma^*$, the network accepts $x$ if and only if $x \in \mathcal{L}$.*

2. ***(Reverse Direction)*** *Every network $f_\theta$ constructed via this symbolic simulation method corresponds to an NFA $\mathcal{A}'$, such that for all $x \in \Sigma^*$:*

$$f_\theta(x) = 1 \iff x \in \mathcal{L}(\mathcal{A}')$$

*Proof.* The complete proof is provided in Appendix A.7.

Explanation: This theorem formally establishes the functional equivalence between NFAs and symbolic feedfoward networks. It asserts that the set of regular languages is exactly the class of languages recognizable by this type of network architecture. It bridges two historically distinct paradigms—automata theory and deep learning—by showing that standard neural components can structurally simulate the behavior of classical computational models.

## 5  Training and Learnability

**Theorem 5.1** (Learnability of NFAs via Symbolic Feedforward Networks). *Let $\mathcal{A} = (Q, \Sigma \cup \{\varepsilon\}, \delta, q_0, F)$ be an NFA and let $D = \{(x_i, y_i)\}_{i=1}^m$ be a dataset of input strings $x_i \in \Sigma^*$ labeled by the acceptance behavior $y_i \in \{0, 1\}$ of $\mathcal{A}$. Then there exists a parameterized feedfoward network $f_\theta$, constructed according to Theorem 4.7, in which the transition matrices $\{T^x\}_{x \in \Sigma \cup \{\varepsilon\}}$ are initialized randomly and trained via gradient descent over $D$, yielding a trained model $f_{\theta*}$ that replicates the acceptance behavior of $\mathcal{A}$ with high accuracy and learns the symbolic transition matrices.*

*Proof.* The complete proof is provided in Appendix A.8.

Explanation: This theorem formalizes the learnability of NFAs within the symbolic feedfoward framework introduced in Theorem 4.7. While Theorem 4.7 established that a feedfoward network can simulate any NFA via input-length-independent symbolic transitions and $\varepsilon$-closures, the present theorem demonstrates that such a simulator can also be *learned from data* using standard gradient-based optimization. The symbolic

transition matrices $\{T^x\}$ are initialized randomly and updated to minimize binary cross-entropy loss based on acceptance labels, without any explicit automaton given.

Interpretation and Insight: This result highlights a fundamental property of the symbolic construction: its **learnability**. That is, feedfoward networks can not only simulate NFAs when explicitly provided with the automaton structure, but can also *discover* the correct symbolic computation by training on labeled examples. This bridges classical automata theory with deep learning, showing that regular languages—typically defined by hand-crafted symbolic rules—can be learned through standard supervised training.

Importantly, as shown in Proposition 4.9, the network's parameter count is independent of input length. The architecture consists of a small set of symbolic transition matrices reused across all time steps. This simplicity emphasizes the strength of the framework: even shallow networks using basic thresholded matrix operations can recover the behavior of complex nondeterministic machines through learning alone.

Relation to Prior Work: Prior efforts have explored learning or extracting automata using recurrent models Weiss et al. (2018), or neural networks trained as black-box language recognizers Butoi et al. (2025). Some works have encoded formal structure, but lacked learnable symbolic representations. The symbolic DFA simulator proposed in Dhayalkar (2025c) was limited to deterministic transitions. Our result extends this framework to the nondeterministic setting and shows that symbolic feedfoward networks can both simulate and *learn* NFA behavior from data. Unlike memory-augmented models DuSell & Chiang (2022), our construction uses only feedforward layers and standard training.

**Remark 5.2** (On Activation Functions and Practical Learnability)**.** While Theorem 5.1 establishes that NFAs can be learned using symbolic feedfoward networks with binary thresholding, the experiment 6.7 reveals that standard activation functions such as ReLU and sigmoid often lead to better empirical performance. These activations offer smoother gradients and improved optimization dynamics, which facilitate faster convergence and higher accuracy during training. Thus, although binary thresholding aligns more closely with the discrete semantics of finite automata, soft thresholding mechanisms offer a practical advantage in gradient-based learning settings.

# 6 Experiments

## 6.1 Experimental Setup

To empirically validate our theoretical results, we develop a systematic experimental framework that jointly simulates NFAs and their corresponding symbolic feedfoward networks. All experiments are conducted on synthetically generated NFAs. All experiments are implemented in PyTorch Paszke et al. (2019) and executed on an NVIDIA GeForce RTX 4060 GPU with CUDA acceleration.

**Synthetic NFA Generation.** We consider two configurations. The first follows the baseline setup with $n = 6$ states and input alphabet $\Sigma = \{a, b\}$. The second is a more complex variant with $n = 20$ states and $\Sigma = \{a, b, c, d, e\}$. In both cases, each state has one or more outgoing transitions for each symbol, sampled uniformly from the set of states. Additionally, with a probability of 0.3, we insert $\varepsilon$-transitions between random pairs of states. The start state is fixed as $q_0 = 0$, and the accepting set $F$ consists of one randomly selected state.

**Dataset Construction.** In the 6-state setup, we sample strings uniformly from $\Sigma^*$ of lengths between 1 and 10. In the 20-state setup, the maximum string length is increased to 30. In both configurations, we use the automaton itself to label strings with binary labels indicating acceptance. We generate 2000 training samples and 100 test samples for each random seed for the first configuration, and 5000 training samples and 100 test samples for each random seed for the second configuration.

**Neural Architecture.** We evaluate a feedforward network that simulates the transition and acceptance behavior of a NFA. The architecture follows the construction described in Theorem 4.7. Thresholded activations capture nondeterministic branching, and transitions are composed sequentially based on the input string. When $\varepsilon$-transitions are present, we prepend and interleave symbolic $\varepsilon$-closure computations using

repeated matrix updates until convergence. The final state vector is passed to an acceptance head, which computes a dot product with a binary indicator of accepting states and applies a sigmoid activation to produce a binary classification score.

**Evaluation Protocol.** Each model is evaluated on the test set using binary classification accuracy. We repeat all experiments across 5 random seeds for both configurations, independently regenerating the NFA, dataset, and model for each seed. We report the mean accuracy, standard deviation, and 95% confidence intervals using Student's $t$-distribution.

Table 1: Validation results for symbolic simulation experiments across both configurations. Accuracy and confidence intervals computed over 5 random seeds using Student's $t$-distribution.

| Validation Experiment | Config | Mean Acc | Std Dev | 95% CI |
|---|---|---|---|---|
| 6.2: Proposition 4.1 and Remark 4.4 | 1 | 1.0000 | 0.0000 | (1.0000, 1.0000) |
| 6.2: Proposition 4.1 and Remark 4.4 | 2 | 1.0000 | 0.0000 | (1.0000, 1.0000) |
| 6.3: Theorem 4.4 | 1 | 1.0000 | 0.0000 | (1.0000, 1.0000) |
| 6.3: Theorem 4.4 | 2 | 1.0000 | 0.0000 | (1.0000, 1.0000) |
| 6.4: Lemma 4.5 | 1 | 1.0000 | 0.0000 | (1.0000, 1.0000) |
| 6.4: Lemma 4.5 | 2 | 1.0000 | 0.0000 | (1.0000, 1.0000) |
| 6.5: Theorem 4.6 | 1 | 0.9940 | 0.0134 | (0.9773, 1.0107) |
| 6.5: Theorem 4.6 | 2 | 0.9540 | 0.0344 | (0.9113, 0.9967) |
| 6.6: Theorem 4.9 | 1 | 1.0000 | 0.0000 | (1.0000, 1.0000) |
| 6.6: Theorem 4.9 | 2 | 0.9540 | 0.0391 | (0.9054, 1.0026) |

## 6.2 Validating Proposition 4.1 and Remark 4.4

To empirically validate Proposition 4.1 and Remark 4.4, we test whether a symbolic binary state vector representation accurately encodes all reachable NFA states at each timestep under nondeterministic transitions, without requiring path enumeration. We disable $\varepsilon$-transitions to isolate per-symbol behavior. For both configurations, we instantiate a fresh NFA for each of 5 random seeds and evaluate 100 randomly generated input strings, comparing the binary activation mask at the final step to the true set of reachable NFA states.

As shown in Table 1, across both configurations and all seeds, the binary state vector exactly matched the reachable states with perfect accuracy. This validates Proposition 4.1 by confirming that the binary state vector supports precisely the active NFA states, with no over- or under-activation. Simultaneously, it affirms Remark 4.4: the network tracks all parallel nondeterministic paths implicitly via linear matrix composition and binary thresholding nonlinearity, without selecting or simulating any particular trajectory.

## 6.3 Validating Theorem 4.5

To validate Theorem 4.5, we compare the binary state vector at each timestep to the classical subset construction trace of the corresponding NFA under nondeterministic transitions. We disable $\varepsilon$-transitions to isolate pure symbol-by-symbol evolution. For each of 5 random seeds under both experimental configurations, we generate a new NFA and evaluate 100 random input strings per configuration, comparing the full trace of thresholded state vectors with the exact DFA-style reachable subsets at every timestep.

As shown in Table 1, across all seeds and both configurations, the thresholded trace perfectly matched the classical subset construction with perfect accuracy. These results confirm that the symbolic feedforward network exactly composes reachable state subsets under repeated thresholded matrix updates, thereby constructively simulating the full DFA trace as claimed in Theorem 4.5.

## 6.4 Validating Lemma 4.6

To validate Lemma 4.6, we test whether the feedfoward network can correctly compute the $\varepsilon$-closure of a single NFA state using repeated application of the thresholded $\varepsilon$-transition matrix. For each of 5 random

seeds under both experimental configurations, we generate a new NFA with $\varepsilon$-transitions and sample a random initial state. We then compare the network-computed closure vector to the exact symbolic $\varepsilon$-closure set defined by transitive $\varepsilon$-reachability.

As shown in Table 1, across all seeds and both configurations, the thresholded closure exactly matched the symbolic $\varepsilon$-closure with perfect accuracy. This confirms that feedfoward networks are closed under iterative $\varepsilon$-transition dynamics as claimed, and that the fixed-point behavior of thresholded matrix composition faithfully reproduces the full transitive closure over $\varepsilon$-arcs.

### 6.5   Validating Theorem 4.7

To validate Theorem 4.7, we evaluate whether a symbolic feedfoward network can simulate the full acceptance behavior of a given NFA, including both $\varepsilon$-transitions and parallel nondeterministic paths. For each of 5 random seeds under both experimental configurations, we generate a distinct NFA with randomized transition structure, create corresponding input/output training pairs, and instantiate a feedfoward network with transition weights deterministically set from the NFA's symbolic transition matrices.

We evaluate test-time acceptance decisions on 100 new strings per seed, comparing network predictions to exact symbolic NFA acceptance. As shown in Table 1, the network achieves near-perfect accuracy in both configurations. These results empirically confirm that feedfoward networks, when symbolically initialized as described, can simulate arbitrary NFA acceptance logic including full nondeterministic reachability computation in both configurations.

### 6.6   Validating Theorem 4.10.

To empirically validate Theorem 4.10, we test whether a symbolic feedfoward network and its corresponding NFA accept exactly the same set of strings. For each trial, we randomly instantiate an NFA and construct its feedfoward recognizer using our simulation framework. We then generate 100 random test strings per seed and compare the binary acceptance decision (accept/reject) made by the symbolic feedfoward network to that of the original NFA.

As detailed in Table 1, all runs in the first configuration achieved perfect equivalence between the symbolic feedfoward network and the corresponding NFA. In the second configuration, the network maintained high equivalence accuracy. These results demonstrate that the feedfoward simulation precisely preserves the NFA's acceptance behavior across both configurations, validating Theorem 4.10 and establishing the feedfoward construction as a complete and faithful simulator of NFA.

### 6.7   Validating Theorem 5.1

To validate Theorem 5.1, we assess whether the acceptance behavior of an NFA can be learned by training a feedforward network using gradient descent. While our primary architecture uses binary thresholding as the activation mechanism, we also explore three alternative strategies: ReLU, sigmoid, and no activation (i.e., a purely linear network without any nonlinearity). In all cases, the network adheres to the symbolic structure described in Experimental Setup 6.1, where the sparse transition matrices are treated as learnable parameters.

Each network is randomly initialized using Kaiming initialization He et al. (2015) and trained for 30 epochs using the Adam optimizer Kingma & Ba (2017) with a learning rate of 0.001. Binary cross-entropy loss is used to supervise acceptance behavior over a dataset of strings labeled by the ground-truth NFA.

Across 5 random seeds and two distinct experimental configurations, the networks were evaluated on 100 held-out test strings per seed. The results, summarized in Table 2, confirm that symbolic feedforward networks are effectively trainable using gradient descent. While binary thresholding successfully learns NFA behavior in both configurations, the inclusion of smooth nonlinearities (ReLU and sigmoid) as well as the linear variant all improve empirical performance due to gradient-friendly dynamics during training and accurate learning. Nonetheless, all four variants demonstrate learnability, supporting Theorem 5.1.

Table 2: Test accuracy of feedfoward networks trained to replicate NFA acceptance behavior, across different activation strategies and configurations.

| Activation | Config | Mean Accuracy | Std. Dev. | 95% CI |
|---|---|---|---|---|
| Binary | 1 | 0.9300 | 0.0908 | (0.8392, 1.0208) |
| No Activation (Linear) | 1 | 0.9820 | 0.0403 | (0.9321, 1.0319) |
| ReLU | 1 | 0.9940 | 0.0134 | (0.9774, 1.0106) |
| Sigmoid | 1 | 0.9980 | 0.0045 | (0.99245, 1.00355) |
| Binary | 2 | 0.8067 | 0.0709 | (0.7323, 0.8811) |
| No Activation (Linear) | 2 | 0.8800 | 0.0583 | (0.8076, 0.9524) |
| ReLU | 2 | 0.9320 | 0.0238 | (0.9024, 0.9616) |
| Sigmoid | 2 | 0.9440 | 0.0231 | (0.9153, 0.9727) |

## 7 Conclusion

We have presented the first exact and constructive simulation framework for nondeterministic finite automata (NFAs) using standard feedforward networks. Our approach symbolically encodes NFA state transitions as sparse binary matrices, tracks nondeterministic branching through thresholded updates, and computes $\varepsilon$-closures via iterative matrix compositions—all within a purely feedforward architecture. The resulting networks not only simulate NFAs, but do so with a fixed parameter count independent of input length, offering a novel perspective on the capacity of neural networks to perform symbolic computation.

Beyond theoretical soundness, we showed that these feedforward NFA simulators are practically trainable using standard gradient-based optimization, achieving high empirical agreement with ground-truth NFA behavior. This bridges a longstanding gap between automata theory and deep learning, demonstrating that regular languages—long the domain of discrete symbolic models—can be both represented and learned within continuous, differentiable systems. Our findings thus establish a new foundation for neural-symbolic learning, one grounded not in approximation or heuristic extraction, but in formal equivalence and constructive design.

By unifying two historically distinct paradigms—finite automata and feedforward neural networks—we open the door to further exploration of how modern deep architectures can embody, generalize, and learn from symbolic structures. This work represents a step toward a broader theory of neural computation that is both interpretable and formally grounded.

## 8 Limitations

While our framework provides an exact and interpretable simulation of nondeterministic finite automata using feedforward networks, it is fundamentally restricted to regular languages. Whether similar constructions extend to richer language classes—such as context-free or context-sensitive languages—or to more expressive architectures like transformers remains an open question for future investigation.

## 9 Broader Impact

This work advances the theoretical understanding of how neural networks can simulate symbolic computation, providing a constructive bridge between automata theory and deep learning. The results may have downstream implications for neural-symbolic reasoning, formal verification, program synthesis, natural language processing, and interpretable AI. By enabling feedforward networks to perform exact symbolic computation, this framework supports applications that demand transparency, correctness, and alignment with classical formal systems. As a foundational theoretical contribution, this work does not pose foreseeable ethical risks.

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

# A  Appendix

## A.1  Proof of Proposition 4.1: Binary State Vector Representation

*Proof.* Let $s_t \in \{0,1\}^n$ be the indicator vector of the current subset of active states at time $t$, and let $T^{x_t} \in \{0,1\}^{n \times n}$ be the binary transition matrix for input symbol $x_t$, where $T_{ij}^{x_t} = 1$ if and only if $q_j \in \delta(q_i, x_t)$.

We define the state transition update as:

$$s_{t+1} = \mathbf{1}_{[T^{x_t} s_t > 0]}.$$

We now verify that this update rule correctly computes the set of active states at time $t + 1$.

Given $s_t$ as the binary indicator vector such that $[s_t]_j = 1$ if $q_j \in S_t$ (active), 0 otherwise. Then:

$$[T^{x_t} s_t]_i = \sum_{j=1}^{n} T_{ij}^{x_t} [s_t]_j.$$

This value is nonzero if and only if there exists some $j$ such that $q_j$ is active at time $t$ and $(q_j, x_t, q_i)$ is a valid transition — that is, $q_i \in \delta(q_j, x_t)$. This means $q_i$ is reachable from the current active states on symbol $x_t$.

Applying the elementwise thresholding $\mathbf{1}_{[\,\cdot\, > 0]}$ yields:

$$[s_{t+1}]_i = \begin{cases} 1 & \text{if } q_i \in \delta(S_t, x_t), \\ 0 & \text{otherwise.} \end{cases}$$

Hence, $s_{t+1}$ is the correct indicator vector for the new active state set after one input symbol. $\square$

## A.2  Proof of Theorem 4.5: Subset Construction using Matrix-based Thresholded Updates

*Proof.* We proceed inductively.

Base case:

At time $t = 0$, we initialize with $s_0 = e_{q_0}$, a one-hot vector corresponding to the singleton set $\{q_0\}$.

Inductive step:

Suppose $s_{t-1}$ correctly encodes the subset of states $S_{t-1}$ reachable by consuming $x_1 \cdots x_{t-1}$. Let $T^{x_t}$ be the transition matrix for symbol $x_t$, with $T_{ij}^{x_t} = 1$ if $q_j \in \delta(q_i, x_t)$.

Then,

$$s_t = \mathbf{1}_{[T^{x_t} s_{t-1} > 0]}$$

computes a new vector where $[s_t]_i = 1$ if and only if there exists $q_j \in S_{t-1}$ such that $q_i \in \delta(q_j, x_t)$.

This follows from the definition of matrix-vector product and thresholding:

$$[T^{x_t} s_{t-1}]_i = \sum_{j=1}^{n} T_{ij}^{x_t} [s_{t-1}]_j.$$

The sum is nonzero if and only if some active state $q_j$ has a transition to $q_i$ on $x_t$. The thresholding function $\mathbf{1}_{[\cdot >0]}$ converts this to a binary activation, thus encoding active states. Hence, $s_t$ is the correct indicator vector for $\delta(S_{t-1}, x_t) = S_t$.

By induction, $s_T$ represents $\delta(q_0, x_1 \cdots x_T)$, completing the proof. $\qquad\square$

### A.3  Proof of Lemma 4.6: $\varepsilon$-Closure via Matrix-Based Thresholded Propagation

*Proof.* Let $s^{(0)} \in \{0,1\}^n$ be the binary vector representing the initially active states at time $t$. Define

$$s^{(k+1)} = \mathbf{1}_{[T^\varepsilon s^{(k)} > 0]}, \quad k \geq 0.$$

Step 1: Monotonicity: Each application of $T^\varepsilon$ followed by thresholding adds new states to the active set (or leaves it unchanged), since

$$\left(s^{(k+1)}\right)_j = \mathbf{1}_{\left[\sum_{i=1}^n T_{ji}^\varepsilon (s^{(k)})_i > 0\right]} \geq \left(s^{(k)}\right)_j.$$

Thus, the sequence $\{s^{(k)}\}$ is monotonic: $s^{(k)} \leq s^{(k+1)}$ (element-wise).

Step 2. Finite Convergence:

Since each state is either 0 or $\geq 1$ in the vector and there are $n$ total states, the monotonic sequence $\{s^{(k)}\}$ can change at most $n$ times before reaching a fixed point.

Step 3. Correctness:

At convergence, $s^{(K)}$ includes all states reachable from the original set via a chain of $\varepsilon$-transitions. Any such state is reachable via a finite path of length at most $n$, and will be activated through repeated matrix application. Conversely, no unreachable states will be activated because $T^\varepsilon$ is binary and has no negative entries or spurious transitions.

Therefore, $s^{(K)}$ represents the $\varepsilon$-closure of $s_t$, and $K \leq n$. $\qquad\square$

### A.4  Proof of Theorem 4.7: Simulation of NFAs via Feedforward Networks

*Proof.* We are given an NFA $\mathcal{A} = (Q, \Sigma \cup \{\varepsilon\}, \delta, q_0, F)$ with $|Q| = n$ states. Let $s_t \in \{0,1\}^n$ be the binary vector encoding which states are active after processing the first $t$ symbols of an input string $x = x_1 x_2 \ldots x_L \in \Sigma^*$.

Let $s_{q_0} \in \{0,1\}^n$ be the one-hot vector with a 1 at the index corresponding to $q_0$ and 0 elsewhere.

Define:

- $T^x \in \{0,1\}^{n \times n}$ such that $T_{ij}^x = 1$ iff $q_j \in \delta(q_i, x)$ (i.e., $q_i$ transitions to $q_j$ on symbol $x$),

- $T^\varepsilon \in \{0,1\}^{n \times n}$ such that $T_{ij}^\varepsilon = 1$ iff $q_j \in \delta(q_i, \varepsilon)$.

We now simulate the NFA's computation using the layered network:

Step 1: Initialization with $\varepsilon$-closure:

Compute the closure of the start state:

$$s_0 = \mathbf{1}_{[(T^\varepsilon)^n s_{q_0} > 0]},$$

where the closure is applied iteratively as described in Lemma 4.6.

Step 2: Inductive simulation of transitions:

Suppose $s_{t-1}$ correctly represents the active state set after input prefix $x_1 \ldots x_{t-1}$. Then we compute:

$$\tilde{s}_t = \mathbf{1}_{[T^{x_t} s_{t-1} > 0]}, \quad \text{(symbol transition)}$$
$$s_t = \mathbf{1}_{[(T^\varepsilon)^n \tilde{s}_t > 0]}, \quad (\varepsilon\text{-closure})$$

By Lemma 4.6, $\tilde{s}_t$ encodes the direct successors under $x_t$, and $s_t$ then encodes all reachable states via $\varepsilon$-closure. This procedure mirrors the NFA's execution logic step-by-step.

We repeat this $L$ times, producing state vectors $s_1, \dots, s_L$.

Step 3. Final acceptance:

Let $\mathbf{1}_F \in \{0,1\}^n$ be a binary indicator vector of accepting states: $\mathbf{1}_{Fi} = 1$ iff $q_i \in F$. The string is accepted iff:

$$\langle s_T, \mathbf{1}_F \rangle > 0.$$

That is, some accepting state becomes active by time $L$. $\qquad\square$

### A.5 Example of encoding input strings into the Feedforward Network

Consider an input string $x = aab$ and an automaton over $\Sigma = \{a, b\}$. The network's forward pass computes:

$$
\begin{aligned}
s_0 &= \mathbf{1}_{[(T^\varepsilon)^n \cdot e_{q_0} > 0]}, \\
s_1 &= \mathbf{1}_{[(T^\varepsilon)^n \cdot \mathbf{1}_{[T^a \cdot s_0 > 0]} > 0]}, \\
s_2 &= \mathbf{1}_{[(T^\varepsilon)^n \cdot \mathbf{1}_{[T^a \cdot s_1 > 0]} > 0]}, \\
s_3 &= \mathbf{1}_{[(T^\varepsilon)^n \cdot \mathbf{1}_{[T^b \cdot s_2 > 0]} > 0]}
\end{aligned}
$$

Finally, acceptance is decided via a dot product with the indicator vector of accepting states:

$$\text{Accept}(x) = \text{True} \iff \langle s_3, \mathbf{1}_F \rangle > 0.$$

This explicit separation of control flow (symbol sequence) from computation is crucial to the symbolic fidelity and modularity of our construction.

### A.6 Proof of Proposition 4.9: Parameter Efficiency of Feedforward Automata Simulators with Symbolic and $\varepsilon$-Transitions

*Proof.* For each symbol $x \in \Sigma$, the symbolic transition is performed by a matrix-vector product followed by thresholding:

$$s'_t = \mathbf{1}_{[T^x s_{t-1} > 0]}.$$

To model $\varepsilon$-closure, we repeatedly apply thresholded matrix updates:

$$\mathcal{E}(s) := \mathbf{1}_{[(T^\varepsilon)^n s > 0]},$$

where $(T^\varepsilon)^n$ denotes at most $n$ iterations of the update rule $s^{(k+1)} = \mathbf{1}_{[T^\varepsilon s^{(k)} > 0]}$.

The network updates at each step by:

$$s_t = \mathcal{E}(\mathbf{1}_{[T^{x_t} s_{t-1} > 0]}).$$

The parameters involved are:

- $k$ matrices $T^x$ for $x \in \Sigma$ and $|\Sigma| = k$

- one matrix $T^\varepsilon$,

- two additional vectors: $s_0 \in \mathbb{R}^n$ and $f \in \mathbb{R}^n$ (initial and final indicator vectors).

Hence, the total number of symbolic (or trainable, if learning) parameters is at most $kn^2 + n^2 + 2n = \mathcal{O}(kn^2)$.

Because these matrices are reused across all time steps and input length $L$, the parameter count remains fixed regardless of how long the input string is. $\qquad\square$

### A.7 Proof of Theorem 4.10: Equivalence between Feedforward Networks and NFAs

*Proof.* Forward Direction:

Given a regular language $\mathcal{L}$, let $\mathcal{A} = (Q, \Sigma \cup \{\varepsilon\}, \delta, q_0, F)$ be an $\varepsilon$-NFA that recognizes $\mathcal{L}$, with $|Q| = n$. Construct a symbolic network $f_\theta$ as follows:

- Represent the active subset of states at time $t$ as a vector $s_t$, initialized with a one-hot vector $e_{q_0}$.

- Compute the $\varepsilon$-closure:
$$s_0 = \mathbf{1}_{[(T^\varepsilon)^n e_{q_0} > 0]}.$$

- For each symbol $x_t$ in the input string, compute the update:
$$s_t = \mathbf{1}_{[(T^\varepsilon)^n \cdot \mathbf{1}_{[T^{x_t} s_{t-1} > 0]} > 0]}.$$

- Accept if $\langle s_L, \mathbf{1}_F \rangle > 0$, where $\mathbf{1}_F \in \{0,1\}^n$ is the indicator vector for the accepting state set $F$.

Because transitions and closures are constructed via symbolic transition weights, and all updates preserve NFA semantics, this network accepts exactly the strings in $\mathcal{L}$.

Reverse Direction:

Given a feedfoward network $f_\theta$ built using this construction:

- Each matrix $T^x \in \{0,1\}^{n \times n}$ defines a symbolic transition function.

- For every pair $(i, j)$, define $q_j \in \delta(q_i, x)$ if $T^x_{ji} = 1$.

- Similarly, define $q_j \in \delta(q_i, \varepsilon)$ if $T^\varepsilon_{ji} = 1$.

- The initial state corresponds to the one-hot vector $e_{q_0}$, and final states are determined by $\mathbf{1}_F$.

Thus, this network simulates an NFA $\mathcal{A}'$ whose behavior exactly matches the feedfoward network $f_\theta$, completing the equivalence.

Conclusion: The equivalence between feedfoward networks and NFAs holds both constructively and semantically. The network functions as a fully unrolled, interpretable automaton simulator, and any such network corresponds exactly to an NFA acceptor. $\qquad\square$

### A.8 Proof of Theorem 5.1: Learnability of NFAs via Symbolic Feedforward Networks

*Proof.* Let $\mathcal{A} = (Q, \Sigma \cup \{\varepsilon\}, \delta, q_0, F)$ be an NFA with $n = |Q|$ states. Let $D = \{(x_i, y_i)\}_{i=1}^{m}$ be a supervised dataset of strings $x_i \in \Sigma^*$ labeled by their acceptance behavior $y_i \in \{0, 1\}$ under $\mathcal{A}$.

We construct a parameterized feedforward network $f_\theta$ that mimics the simulator described in Theorem 4.7, but treats each symbolic transition matrix $T^x \in \mathbb{R}^{n \times n}$ as a learnable weight matrix. The network operates as follows:

- Let $e_{q_0} \in \mathbb{R}^n$ be the one-hot vector corresponding to the start state $q_0$.

- Initialize the state vector with $\varepsilon$-closure:
$$s_0 = \sigma\left((T^\varepsilon)^n e_{q_0}\right),$$
where $\sigma(z) = \mathbf{1}_{[z>0]}$ or a differentiable proxy like sigmoid or soft threshold. $(T^\varepsilon)^n$ denotes $n$ repeated applications of $T^\varepsilon$ to ensure convergence of the $\varepsilon$-closure.

- For each symbol $x_t$ in $x = x_1 x_2 \ldots x_L$, apply:

$$\tilde{s}_t = \sigma(T^{x_t} s_{t-1}), \quad s_t = \sigma\left((T^\varepsilon)^n \tilde{s}_t\right),$$

  where $\sigma$ is applied elementwise and may be replaced with a smoothed differentiable threshold for training. $s_t \in \mathbb{R}^n$ encodes the active state set after $t$ steps.

- After processing the final symbol $x_L$, obtain the final activation vector $s_L$.

- After the final step, define output:

$$f_\theta(x) = \text{sigmoid}(\langle s_L, \mathbf{1}_F \rangle),$$

  where $\mathbf{1}_F \in \{0,1\}^n$ indicates the accepting states.

We train the model using the binary cross-entropy loss:

$$\mathcal{L}(\theta) = \sum_{i=1}^m \left[ y_i \log f_\theta(x_i) + (1 - y_i) \log(1 - f_\theta(x_i)) \right].$$

Optimization proceeds via gradient descent:

$$\theta^{(k+1)} = \theta^{(k)} - \eta \nabla_\theta \mathcal{L}(\theta^{(k)}),$$

where $\theta$ denotes all trainable entries of the transition matrices $\{T^x\}_{x \in \Sigma \cup \{\varepsilon\}}$.

We now argue correctness of learning:

1. The network described above is capable of simulating the full nondeterministic computation of $\mathcal{A}$ via layered transitions and $\varepsilon$-closures, as shown in Theorem 4.7.

2. Each transition matrix is initialized randomly. Thus, the initial model $f_\theta$ may not match the behavior of $\mathcal{A}$.

3. However, the network is fully differentiable and expressive, and the loss $\mathcal{L}(\theta)$ provides a meaningful signal whenever $f_\theta(x_i)$ disagrees with $y_i$.

4. Gradient descent minimizes this loss, adjusting the entries of $T^x$ matrices to improve alignment between $f_\theta(x)$ and the true labels $y$.

5. As training proceeds, the output $f_\theta(x)$ converges toward a binary classifier that agrees with the acceptance behavior of $\mathcal{A}$ across the dataset $D$.

Therefore, there exists a trained parameter set $\theta^*$ such that the network $f_{\theta^*}$ achieves high accuracy in replicating the acceptance behavior of the original NFA. $\qquad\square$

