# OpenReview forum: "Neural Networks as Universal Finite-State Machines: A Constructive Feedforward Simulation Framework for NFAs"
_TMLR — Rejected by TMLR_

### Review · Reviewer_s4d1 · 2025-06-05

**Summary Of Contributions:**

This paper adresses the topic of connectig formal language / automata theory with neural networks. Thereby, the paper claims to show an equivalence between classical nondeterministic finite automata and feedforward neural networks with ReLU activations.

**Audience:**

No

**Broader Impact Concerns:**

no concerns

**Claims And Evidence:**

No

**Requested Changes:**

I can only recommend a complete revision of the manuscript, with particular emphasis on establishing a clear and rigorous definitional foundation and aligning the presentation with basic scientific standards.

**Strengths And Weaknesses:**

This paper does not meet the standards expected of a scientific publication and, in its current form, should not be considered for acceptance.

While the topic is undoubtedly intriguing and the paper appears polished at first glance, it suffers from several critical shortcomings:

- The notion of equivalence between NFAs and FNNs, which underpins several key claims, is never formally defined in the paper.
- The presentation style is highly unorthodox, with theorems, lemmas, and other results embedded as subsection titles, which detracts from the readability and scientific rigor of the manuscript.
- Most formal results are either (1) trivial (e.g., Theorem 1), or (2) highly underspecified and argued at an overly abstract level (e.g., Lemma 3), with little to no technical depth.
- Some statements are factually incorrect. For instance, Corollary 1, titled "ReLU acceptors are NFA-complete", makes a number of problematic statements. First, the notion of a ReLU acceptor is never properly introduced. Second, the term NFA-complete is highly unusual and not standard in the literature. The corollary claims that a 3-layer ReLU network can simulate any NFA, but this is misleading at best. Even if we momentarily ignore the undefined semantics of what it means for an FNN to “accept” a regular language, the proof (Appendix A.6) indicates that the network depth depends on the input length, stating a depth of O(T+2n), where T is the length of the input. This implies that the construction does not yield a single fixed network per NFA, especially not one with a fixed number of layers, but rather one network per input length. Assuming that ambiguities are resolved, this is also straightforward.

---

> ### Author Response · Authors · 2025-08-04
>
> We thank the reviewer for their detailed comments. We appreciate the time and effort taken to evaluate our work and have made substantial improvements in response to the concerns raised. Below, we address each point raised in the review.
>
> - Lack of formal definition of equivalence between NFAs and FNNs:
> Addressed. We have now included a rigorous and unambiguous formal definition of equivalence in Definition 3.7 of the revised manuscript. This clears up any ambiguity and establishes the exact semantic conditions under which our simulation is said to hold.
>
> - Unorthodox presentation style (theorems/lemmas embedded in section headers)
> Addressed. The revised paper adopts a more conventional academic structure. Theorems, lemmas, and corollaries are now presented as clearly numbered blocks within the main body of the text, rather than as section titles, aligning with standard formatting.
>
> - Triviality and lack of technical depth in some results
> Addressed. We have carefully audited all results for substantive contribution and technical depth. Simple or foundational results (e.g., what was previously Theorem 1) have been moved to propositions or lemmas with clear motivation, and more significant theorems now include complete formal proofs in the appendix. The body of the paper highlights the key insights and innovations behind each result
>
> - Factual inaccuracies and terminological issues (e.g., "ReLU acceptors", "NFA-complete", variable depth)
> Clarified and Corrected. We have removed the phrase “NFA-complete,” acknowledging that it is non-standard and potentially misleading. The term ReLU acceptors is now removed. Instead, we now use the terminology of "equivalence" and have formally defined it in Definition 3.7.
> Regarding depth: it is indeed true that the number of layers depends on the input length T. We have clarified this explicitly in Theorem 4.6 and Proposition 4.8, which emphasize that while the depth scales with input length, the number of distinct trainable parameters is fixed and independent of it. This distinction is crucial and aligns with the classical notion of computation unrolling in automata simulation.

---

> > ### Author Response · Authors · 2025-08-04
> >
> > We have submitted a revised manuscript that incorporates all reviewers feedback. The updated version addresses each suggestion and question in detail and now follows standard formatting conventions for theorems, lemmas, propositions, and remarks. We kindly invite you to review the updated manuscript.

---

### Review · Reviewer_KmYk · 2025-06-16

**Summary Of Contributions:**

This submission claims that nondeterministic finite automata (NFAs) can be simulated by ReLU neural networks. Both theoretical evidence (in the form of a construction in Theorem 3) and empirical evidence on simple NFA datasets are provided.

**Audience:**

No

**Claims And Evidence:**

No

**Requested Changes:**

Due to the incomplete theoretical results and overstated claims, major revisions to fix both of these are required before the submission can be reconsidered.

I recommend the following additional changes to be made:
- (major) Lemma 1 appears to be exactly the same as Theorem 1. How do these differ?
- (major) The clarity and formatting of the paper are lacking. In section 4, each of the main theorems/lemmas should be clearly stated in a theorem environment. Moreover, the experimental results in section 5 should be presented in a plot/table form to be easier to understand.
- (minor) Missing reference on simulating automata with transformers: Transformers learn shortcuts to automata. Bingbin Liu, Jordan T. Ash, Surbhi Goel, Akshay Krishnamurthy, Cyril Zhang. ICLR 2023. In particular, this paper shows how automata can be simulated with a $O(\log T)$ depth transformer.

**Strengths And Weaknesses:**

Strengths:
- It is indeed an interesting question to connect the expressive power of neural networks to computational models such as automata.
- To the best of my knowledge, prior work has focused on deterministic finite automata / Turing machines, and I am not aware of prior work for the nondeterministic setting.

Weaknesses:
I find the main claims of the paper to be incomplete and overstated:
1. The main construction is missing key details on how the input string $x$ is passed into the neural network. In particular, in Lemma 1/Theorem 1, the authors claim that a single ReLU layer is sufficient for computing the transitions after consuming the input symbol $x_t$. However, this construction proceeds by directly encoding the transition function $T^{x_t}$ as the weights of the ReLU layer. No details are provided about how a neural network can take in $x_t$ (say as a one-hot vector in $\mathbb{R}^{|\Sigma|}$) and $s_t$ and map these to $s_{t+1}$.
2. Theorem 3 states that there exists a three-layer ReLU network which can simulate NFAs (this "three-layer" claim is repeated in the abstract and introduction as well). However, the construction for Theorem 3 actually requires $O(nT)$ ReLU layers -- one layer for each token in the input string, and $n$ layers to compute the $\epsilon$-closure, which is repeatedly interleaved between each per-symbol transition layer. This "three-layer" claim must be edited.
3. The paper states that one contribution of their construction is that it does not require recurrence. However, their construction which has depth scale with the input length is essentially equivalent to unrolling a recurrent construction for $T$ steps. As such, I find the claim that this approach does not rely on recurrent mechanisms to be misleading.
4. Theorem 1/Lemma 1 state that $s_t$ is a binary vector. However, for this to be true, the activation function must be the indicator $\mathbf{1}(z \ge 0)$, instead of ReLU. Indeed, this remark is made in Appendix A.1, that only positivity of the entries of $s_t$ matters. But if this is indeed the case, why is the ReLU necessary? This exact same positivity is obtained from having no activation (i.e a linear neural network).
5. The claims on learnability via gradient descent are also overstated. Lemma 3 assumes that the neural network is initialized to match the construction in Theorem 3, and moreover that the "sparsity and structure of transition matrices" are preserved. Firstly, as stated in point (1), an explicit construction for the weights of the neural network are not provided, and so preserving the sparsity of $\delta$ does not seem to make sense (and it is not clear how this would be actually implemented). Secondly, preserving the sparsity is equivalent to already knowing /hardcoding the correct NFA; as such, there is no learning happening. Finally, Theorem 4 implicitly assumes that GD converges to zero population loss -- this seems unreasonable as an assumption, and moreover must be explicitly stated in the theorem.
6. (minor) The $O(T)$-depth construction seems to be quite parameter inefficient; justification must be provided on why such a construction is interesting.

---

> ### Author Response · Authors · 2025-08-04
>
> We thank the reviewer for their thoughtful and detailed comments. We greatly appreciate the recognition of our paper’s originality in addressing the nondeterministic setting, and we acknowledge the areas where clarification and revision were needed. We have revised the manuscript substantially to address all concerns, and we outline our detailed responses below.
>
> - Clarification on input string representation and transition computation
> We fully agree with this observation. In the revised manuscript, we have introduced a new subsection titled "Encoding Input Strings into the Feedforward Network" in Section 4.1 that explicitly clarifies how the input string is processed. Each symbol from the input string acts as a selector, choosing the corresponding transition matrix from a fixed set of matrices. The input string is not embedded or fed into the network as a vector; instead, the network control flow is symbolically driven by the string sequence itself.
>
> - Claim regarding “three-layer” ReLU network
> Thank you for pointing this out. We have revised all claims in the abstract, introduction, and theorems to clarify that the depth of the constructed network is linear in the input length, with fixed parameter count, and shared transition matrices. The phrase “three-layer” has been removed or clarified wherever it occurred. We now emphasize that the depth scales with the input length, but not the number of parameters, which is fixed and independent of the input size.
>
> - Recurrent vs. unrolled view
> We agree that the depth-unrolled architecture can be viewed as an unrolling of a symbolic recurrent mechanism.  However, our contribution lies in providing an exact symbolic feedforward construction that avoids “black-box” recurrence. While the structure resembles recurrence in depth, the network does not learn or store recurrent hidden states, but instead performs symbolic matrix applications controlled by the input sequence.
>
> - Use of binary threshold vs ReLU
> We agree with the reviewer’s concern regarding the use of ReLU in prior versions. In the updated paper, we emphasize that our construction does not require ReLU or any specific activation function. The simulation of NFAs can be achieved using binary thresholding (i.e., an indicator function), and this is the default interpretation in our proofs.
> However, to support practical implementations and learning via gradient descent, we note—explicitly in Remark 4.2, Remark 5.2, and Experiment 6.7—that other activations like ReLU, sigmoid, or even no activation (linear) can also be used without affecting the correctness of the simulation. These alternatives allow smoother optimization landscapes during training while preserving the expressive structure of the network.
> This flexibility strengthens the generality of our construction and bridges theoretical fidelity with empirical trainability.
>
> - On learnability.
> We have rephrased and strengthened our claims with respect to learnability in Theorem 5.1, and experiment 6.7 provides empirical evidence where the network is first randomly initialized using Kaiming Initialization before training.
>
> - Parameter inefficiency in O(nL) depth
> Although the depth scales with the input length $L$, the number of unique parameters (i.e., one matrix per symbol) is fixed and independent of input length. This is explained in Proposition 4.9 in more detail.
>
> - Lemma 1 appears to be exactly the same as Theorem 1. How do these differ?
> State vector representation (updated to Proposition 4.1) precisely defines how the NFA state set is represented as a vector and how the transition operation updates this vector upon reading a symbol. It forms the foundation of our symbolic simulation: each application of a transition matrix to the current state vector produces the next reachable states.
> Separately, Parallel Path Tracking (updated to Remark 4.4) explains how this vector-based representation naturally supports parallel tracking of multiple computational paths—a core characteristic of nondeterministic automata. The vector inherently captures all active states at each step, which allows the network to evolve all possible paths simultaneously. This distinction has been made clear in the current version to eliminate any confusion between the function of the proposition and the accompanying remark.
>
> - Formatting and clarity improvements
> All theoretical results are now numbered and presented using formal theorem/lemma environments. In addition, the experimental results in Section 6 are presented in well-structured tables, with accuracy, standard deviation, and confidence intervals clearly reported.
>
> - Reference to paper Transformers learn shortcuts to automata. Bingbin Liu, Jordan T. Ash, Surbhi Goel, Akshay Krishnamurthy, Cyril Zhang. ICLR 2023 has been added. Thank you for the suggestion

---

> > ### Comment · Reviewer_KmYk · 2025-09-02
> > **Response to Authors**
> >
> > Thank you to the authors for your detailed response.
> >
> > My main concern still lies with how the input string $x$ is passed into the neural network. I appreciate the addition of Section 4.1 which provides more details on the construction. However, this construction is not a standard feedforward network, as is claimed in the abstract. A standard feedforward network would take as input a single vector in $\mathbb{R}^{L|\Sigma|}$ which concatenates the one-hot encodings of each token. The "symbol-driven matrix selection" approach utilized in this paper, which maintains a trainable transition matrix for each symbol in $\Sigma$, and composes them in order of the tokens in the string, is not reminiscent of any standard neural network architecture but rather seems to be a specific parametric model which is explicitly chosen for the NFA simulation task. The architecture it is closest to is an RNN/SSM, but an RNN would still need to take in an embedding of $x_t$ and the state vector $s_t$ and map this tuple to the next state vector $s_{t+1}$; how this can be done with "standard" architectural approaches is not described.
> >
> > Next, I still maintain that the authors' construction relies on recurrence. I do appreciate the clarification that the number of parameters is independent of the input length $L$. However, the architecture used here applies the same function to each token $x_t$ in the input string, the state vector $s_t$ is just a recurrent hidden state, and so in my opinion the architecture considered here is some form of a recurrent model.
> >
> > Finally, the proof of learnability (Theorem 5.1) is not at all rigorous. Claim 4 in Appendix A.8 directly states that gradient descent minimizes the loss. This is not something that can be assumed -- the loss landscape is non-convex, and so you must prove that GD indeed converges to a global minimizer.
> >
> > As such, I recommend rejection.

---

### Review · Reviewer_L4tk · 2025-07-25

**Summary Of Contributions:**

The authors present a constructive framework showing that feedforward ReLU neural networks can exactly simulate any nondeterministic finite automaton (NFA). They show that a 3-layer ReLU network can simulate an NFA with n states using O(n) width. Furthermore, they show that gradient descent, when constrained to preserve the structure of symbolic transition matrices in the network, maintains the NFA's acceptance behavior. Finally, with careful empirical analysis across various symbolic tasks, they show near-perfect alignment between NFAs and their ReLU counterparts.

**Audience:**

Yes

**Broader Impact Concerns:**

I don't have any concerns.

**Claims And Evidence:**

Yes

**Requested Changes:**

Please check my questions above.

**Strengths And Weaknesses:**

The primary strength of the paper lies in its careful analysis and demonstration of the construction to encode NFAs in ReLU networks. As such, the authors provide the first constructive proof that any NFA (and thus any regular language) can be simulated exactly by a shallow, feedforward ReLU network with no recurrence or memory. Furthermore, the authors take a step forward and show that gradient descent training can retain the symbolic semantics of the NFA under structure-preserving constraints. Altogether, the authors present a clear demonstration on how ReLU dynamics can represent all core functions in NFAs. Finally, the authors conduct empirical studies with hand-constructed sparse weight matrices corresponding to the NFA transition functions and show that the ReLU models have near perfect alignment by checking  path tracking, subset construction, $\epsilon$-closure and end-to-end acceptance. Overall, the authors provide comprehensive argument to show the connection between ReLU networks and NFAs.



As such, I don't see any primary weaknesses. I have few questions regarding the theoretical and experimental setup.

- What is the necessary precision to construct the ReLU network? Will precision depend on the parameters of the NFA (e.g. transition matrix values)?

- How robust is the construction to small perturbations in the input or transition matrix? This is connected to gradient descent analysis, as it's unclear how strict the necessary initial condition on the network is and how much noise (and what kind of noise) can be present in gradient descent.


- Have the authors tried training a ReLU network on the strings generated from NFAs? What will be an appropriate loss function to train such a model? Can the authors comment on how easy it is to train such networks? Will it require some careful initialization and how would the size of the model affect the convergence of the model?

---

> ### Author Response · Authors · 2025-08-04
>
> - What is the necessary precision to construct the ReLU network? Will precision depend on the parameters of the NFA (e.g. transition matrix values)?
> Answer:
> The symbolic simulator is not robust to perturbations: since the construction is discrete and exact (i.e., based on 0/1 values for transitions and state vectors), even small perturbations to the transition matrix or input encoding can lead to incorrect transitions. This behavior is expected, as NFAs are themselves symbolic automata with crisp semantics.
> However, when training such a network using gradient descent (Section 6), we rely on relaxed, real-valued versions of the symbolic transition matrices. These real-valued matrices are optimized via gradient descent and can tolerate small perturbations. In this regime, the robustness depends on:
> 1. The margin between correct and incorrect state activations.
> 2. The saturation behavior of the activation function (e.g., sigmoid is smoother, ReLU introduces sparsity).
> 3. The distribution of the data (e.g., how separable the positive and negative strings are in transition space).
> Thus, while the symbolic construction is brittle (by design), the learned variant exhibits smooth robustness to small noise, as supported by our experiments.
>
> - How robust is the construction to small perturbations in the input or transition matrix?
> Answer:
> The symbolic simulator when directly encoded from the known transition function is not robust to perturbations: since the construction is discrete and exact (i.e., based on 0/1 values for transitions and state vectors), even small perturbations to the transition matrix or input encoding can lead to incorrect transitions. This behavior is expected, as NFAs are themselves symbolic automata with crisp semantics.
> However, when training such a network using gradient descent (Section 6), we rely on relaxed, real-valued versions of the symbolic transition matrices. These real-valued matrices are optimized via gradient descent and can tolerate small perturbations. In this regime, the robustness depends on:
> 1. The margin between correct and incorrect state activations.
> 2. The saturation behavior of the activation function (e.g., sigmoid is smoother, ReLU introduces sparsity).
> 3. The distribution of the data (e.g., how separable the positive and negative strings are in transition space).
> Thus, while the symbolic construction is brittle (by design), the learned variant exhibits smooth robustness to small noise, as supported by our experiments.
>
> - Network training and loss function.
> Answer: Yes, as described in Section 6.7, we train networks to learn NFAs from labeled string data. The setup assumes that the ground-truth NFA is unknown, and the network must discover the structure from positive and negative examples. With respect to loss function, we use binary cross-entropy loss to train the model, as described in Theorem 5.1, Section 6.7 and Appendix A.8 Proof of Theorem 5.1. Training the model was relatively straightforward, following standard training and testing procedures for neural networks, and initialization can be carried out using standard random initialization like Kaiming initialization or Xaview initialization

---

> ### Author Response · Authors · 2025-08-04
>
> We have submitted a revised manuscript that incorporates all reviewers feedback. The updated version addresses each suggestion and question in detail and now follows standard formatting conventions for theorems, lemmas, propositions, and remarks. We kindly invite you to review the updated manuscript.

---

> > ### Comment · Reviewer_L4tk · 2025-09-10
> >
> > Hi,
> >
> > Thank you for the detailed response. I carefully went through reviewer KmYk's review and the author's responses. I agree with reviewer KmYK's concerns that the theoretical statements lack rigor. As such, I would like the authors to take reviewer KmYK's concerns into consideration for the next version of the paper.
> >
> > As such, I recommend rejection as well.

---

### Author Response · Authors · 2025-08-04
**Manuscript Revised and ready for review**

To all reviewers,

We have submitted a revised manuscript that incorporates all reviewers feedback. The updated version addresses each suggestion and question in detail and now follows standard formatting conventions for theorems, lemmas, propositions, and remarks. We kindly invite you to review the updated manuscript.

Thank you,
Paper authors

---

### Decision · Action_Editor_AnNS · 2025-09-17

**Recommendation:** Reject

**Additional Comments:**

This paper presents a construction to simulate nondeterministic finite automata (NFAs) using ReLU feedforward network, and further assert that this can be learned from data using standard gradient descent.

While the authors made significant efforts to revise the manuscript based on initial feedback, the reviewers unanimously found that paper failed to address two major concerns:

- Misleading Claims/Lack of Novelty: The paper states that its model is a "standard feedforward network" that avoids recurrence. However, reviewers KmYk and s4d1 strongly argued that the architecture, whose depth scales with input length and reuses parameters, is functionally an unrolled Recurrent Neural Network (RNN). Hence, the core result of the paper reduces to 'RNNs can simulate finite automata' which is a well-established fact dating back to the 1990s.

- Lack of Theoretical Rigor: A critical flaw, highlighted by Reviewer KmYk and later agreed upon by L4tk, is the unrigorous proof of learnability (Theorem 5.1). The proof incorrectly assumes that gradient descent will converge to a global minimum of the loss function without providing any justification.

Given that the paper's central claims are either not novel or lack rigorous proof, the paper is not ready for publication.

**Audience:**

Yes

**Audience Explanation:**

Yes, there is sufficient interest in the community on understanding the representation power of deep networks.

**Claims And Evidence:**

No

**Claims Explanation:**

There were major concerns raised by the reviewers regarding the claims of the paper that I describe in the comments.